# Self-Supervision for Medical Image Classification: State-of-the-Art Performance with ~100 Labeled Training Samples per Class

**DOI:** 10.3390/bioengineering10080895

**Published:** 2023-07-28

**Authors:** Maximilian Nielsen, Laura Wenderoth, Thilo Sentker, René Werner

**Affiliations:** Department of Computational Neuroscience, Institute for Applied Medical Informatics, University Medical Center Hamburg-Eppendorf, Martinistraße 52, 20251 Hamburg, Germany; laura.wenderoth@stud.uke.uni-hamburg.de (L.W.); t.sentker@uke.de (T.S.); r.werner@uke.de (R.W.)

**Keywords:** self-supervision, deep learning, image classification, medical imaging

## Abstract

Is self-supervised deep learning (DL) for medical image analysis already a serious alternative to the de facto standard of end-to-end trained supervised DL? We tackle this question for medical image classification, with a particular focus on one of the currently most limiting factor of the field: the (non-)availability of *labeled* data. Based on three common medical imaging modalities (bone marrow microscopy, gastrointestinal endoscopy, dermoscopy) and publicly available data sets, we analyze the performance of self-supervised DL within the self-distillation with no labels (DINO) framework. After learning an image representation *without* use of image labels, conventional machine learning classifiers are applied. The classifiers are fit using a systematically varied number of labeled data (1–1000 samples per class). Exploiting the learned image representation, we achieve state-of-the-art classification performance for all three imaging modalities and data sets with only a fraction of between 1% and 10% of the available labeled data and about 100 labeled samples per class.

## 1. Introduction

Medical image analysis is currently dominated by supervised deep learning (DL). DL has been shown to achieve exceptional performance for many medical imaging applications and benchmarks [1,2,3], and is sometimes claimed (for specific applications) to be on par with human experts [4]. The key to success is usually a large amount of *labeled* data, i.e., image data and corresponding task-specific expert-annotated labels, that are used for DL model training. While the availability of medical image data is, for routinely acquired images and standard diagnoses, usually not problematic per se, comprehensive labeling of large data sets by medical experts is a major hurdle: It is a time-consuming and costly process that often even requires multiple experts to evaluate the same data due to high inter- and intra-rater variability of clinical scores. For rare diseases, on the contrary, even the availability of a sufficient amount of image data for supervised end-to-end training of DL models is sometimes not given. Thus, the need for a large number of expert-annotated data poses a bottleneck in automated medical image analysis when using the current methodical state-of-the-art approach: supervised deep learning.

Interestingly and despite existing large annotated data sets like ImageNet (currently more than 14 million images, [5]) or the COCO data set (more than 200,000 labeled images, [6]) and impressive performance of supervised DL in corresponding benchmarks, recent developments in the natural image and computer vision domain showed a trend toward self-supervised learning (SSL). Unlike supervised methods, SSL is purely data-driven. It aims to learn a generalized representation of the input directly from the presented data, independent of labels. In practice, this is described to result in better generalizability of SSL models and increased robustness when out-of-distribution samples are present [7,8]. Theoretical foundations and explanations for SSL and the behavior of different SSL variants are part of active research [9]. However, the expectation is that based on the learned generalized and robust input representation, downstream tasks like image classification can be solved more efficiently, i.e., with a reduced amount of ground truth labels [10]. Accordingly, self-supervision potentially allows combining the advantages of DL (extraction of meaningful features from high-dimensional input data like images) and conventional machine learning (ML) methods that offer run-time efficient and robust classification with only a few but well-represented labeled data points.

The SSL concept, therefore, holds the promise to reduce the annotation workload and to accelerate research in medical imaging, but there is more to it than directly meets the eye: While the state-of-the-art for natural image domain benchmarks for image classification steadily progresses with the introduction of new and often larger DL architectures, the performance on many medical image domain benchmarks has stagnated after initial success. The problem is: As the parameters of (DL) models increase, so do the requirements for labeled data, and for some applications and benchmark data sets, current DL models are only trainable with the inclusion of additional large annotated image data sets. For the above reasons, appropriate (i.e., large) data sets are difficult to obtain for the medical image domain. At this, self-supervised DL could help to optimize the performance on limited-size benchmark data sets by exploiting unlabeled data to improve robustness and generalizability of the learned representation. Moreover, the ability to quickly adjust only the classifier of an already fitted pipeline and treat the DL model as a static feature extractor is, from a clinical perspective, particularly interesting: As retraining of conventional ML classifiers is much faster than retraining or adaption of end-to-end DL classification systems, it offers a way for time-efficient integration of additional labeled data or extension to new classes and tasks.

Despite its promising capabilities (see, e.g., current perspectives [11] and reviews [12,13,14]), SSL is still in its infancy in the field of medical image analysis. Related work mainly focused on self-supervised pretraining of DL models, with the models still being fine-tuned on relatively large labeled data sets. Aziz et al., for instance, demonstrated self-supervised pre-training using unlabeled data to improve subsequent medical image classification; the classifier training was, however, still based on >15,000 labeled images [15]. To our knowledge, a systematic experimental analysis of the label data efficiency claim, i.e., the classification performance for limited label data scenarios, on publicly available medical data sets has not been performed.

This study aims to fill this gap. To substantiate the data efficiency claim and to define first related benchmark data, we analyze the potential of SSL for different medical image classification settings and demonstrate for three established but distinct public data sets (cell images obtained from bone marrow smears, endoscopic images, and dermoscopic lesion images) that it is possible to reach (near) state-of-the-art performance with as little as approximately 100 labeled training samples per class.

There exist different approaches to SSL, with contrastive learning approaches like SimCLR [16] being commonly used. Recently, the so-called DINO algorithm (self-*di*stillation with *no* labels) has been introduced, which shifts away from contrastive learning and is based on momentum encoders, i.e., BYOL [17] and siamese learning [18]. Compared to SimCLR, DINO improved the performance on the Imagenet benchmark for SSL approaches by over 6% using the same network architecture (ResNet50) [16,19]. Motivated by this leap in performance, our study also builds on the DINO framework. Methodically, we further deviate from the currently common practice of finetuning a linear layer on top of the SSL-pretrained model or using SSL only as a means of pretraining before fitting the entire model on available annotated data [14]. Instead, conventional ML classifiers are used on top of the SSL-trained models. This allows for an almost instantaneous fitting process and easy adoption to other downstream tasks. Moreover, we provide a highly customizable and adaptive pipeline of our reimplementation of the DINO algorithm using community standard APIs for others to take full advantage of our experiments and findings.

## 2. Methods

The experiments are based on the following publicly available medical image data sets: a bone marrow single cell data set, published by Matek et al. [20]; an endoscopic image data set assembled and released by Borgli et al. [21]; and a dermoscopic lesion data set, released as part of the annual Grand Challenges organized by the International Skin Lesion Collaboration (ISIC) [22,23,24,25,26]. The implementation of the DINO framework, the deep learning models, and our experiments is based on the pytorch and scikit-learn frameworks. The source code, the trained models, and the data split information for the data sets as used in this study are provided publicly available at https://github.com/IPMI-ICNS-UKE/sparsam (accessed on 23 July 2023). Subsequently, the methods and data sets are therefore primarily described conceptually. Details such as hyperparameter values can be found in the source code and the Github repository.

### 2.1. Dino: Knowledge Distillation with No Labels

DINO is a self-supervised DL framework that follows the concept of knowledge distillation. Knowledge distillation refers to the process of knowledge transfer between two models, one often referred to as the teacher and the other as the student model. In DINO, both models have the same architecture but are trained on differently-sized patches of the input image data. The idea is to learn a consistent representation of local views (smaller patches, input to the student model) and global views (larger patches, input to student and teacher models) of the same image. Thus, the models are trained without using any annotations (=labels) of the image data.

This general concept is summarized in Figure 1, following [19]: For an input image *I*, differently sized patches of the image are presented to the two DL models of equal architecture but a different set of model parameters: larger patches (global views) $I_{g}$ to the teacher network (parameterized by $\theta_{t}$) and a smaller patch (local view) $I_{l}$ to the student network (parameterized by $\theta_{s}$). Following the default DINO implementation, for each input *I*, two global (>50% of the image area) and five local views (<50% of the image area) are extracted. All crops are further subject to extensive augmentation (horizontal/vertical flipping, rotation, color jitter, grey scaling, Gaussian blurring, solarization) and define a set $I=I_{g}\cup I_{l}$. The training objective is given by the categorical cross entropy (CE) between student and teacher outputs $P_{t}$, $P_{s}$ evaluated for all combinations of the global views (input to the teacher) and the remaining elements of I (input to the student) after temperature-weighted softmax normalization.

While the student parameters $\theta_{s}$ are optimized by stochastic gradient descent, the teacher parameters $\theta_{t}$ are given as an exponential moving average of $\theta_{s}$. The optimization is further stabilized by a centering of the teacher outputs.

The general DL model architecture used within the DINO framework consists of two major parts: the backbone network and a projection head. As the backbone network, we use a cross-variance vision transformer (XCiT, see Section 2.2). The projection head consists of a series of four fully connected layers as proposed in [19]. During network inference (i.e., final application of the network), only the backbone with the teacher weights $\theta_{t}$ is used to generate the features that are used for the desired downstream task (here: medical image classification).

### 2.2. Xcit: Cross-Covariance Image Transformer

Similar to the original DINO paper [19], we built on a vision transformer as the backbone network. Offering a reasonable trade-off between required computational resources and performance, we used the XCiT network, a recent version of the *Vision Transformer* family, in the *small* variant [27]. Different to the original self-attention as introduced in [28], which operates directly across the tokens, XCiT derives the attention map from the cross covariance matrix of key and query projections of the token features. This results in a linear complexity with regard to the number of tokens [27] and allowed us to work on batch sizes similar to [19]. We did not train the XCiT networks from scratch, but further tuned models pretrained on ImageNet.

**Figure 1 bioengineering-10-00895-f001:**
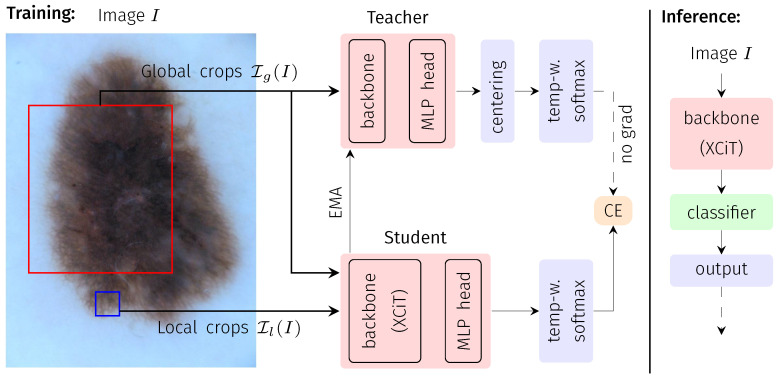
Flowchart of the DINO training and inference processes. During training, an input image is randomly cropped into two global and five local crops, the global crops are passed through the teacher network, and the network outputs are centered (MLP: multi-layer perception). The student network receives the same two global crops but is also presented with the additional five local crops and outputs are subsequently not further post-processed. Student and teacher have the same network architecture, as illustrated. After softmax activation (sharpened by temperature weighting), the loss between student and teacher outputs is calculated in terms of categorical cross entropy (CE). During optimization, only the student is updated via backpropagation, while the teacher weights are an exponential moving average (EMA) of the student weights (i.e., no gradient backpropagation through the teacher). During inference (right block), the trained teacher backbone is directly employed to extract the embedding of the entire image. The corresponding embedding is subsequently fed into a conventional ML classifier.

### 2.3. Image Classification Based on DINO Representation

Image classification was based on the learned representation of the trained DINO backbone (XCiT). We only deployed the teacher model, even though in theory teacher and student should lead to similar performance at convergence. The process of representation extraction and classification is also shown in Figure 1. Before classifier training by means of a small subset of the labeled training images (see Section 2.6 for details), the individual features of the representation of these training images were first re-aligned by a principal component analysis without dimension reduction and subsequently normalized along each feature dimension to a zero mean and unit variance by a logarithmic power transform. These preprocessed features and the corresponding image labels were then used to fit the parameters of the ML classifiers. We applied three standard ML classifiers with default parameters: support vector machine (SVM; kernel: radial basis function), logistic regression (LR), and K-Nearest Neighbors (KNN; K=10). For testing, the representation of the image to be classified was extracted the same way as described above and transformed using the previously fitted PCA and power transform. Then, the trained ML classifier was applied.

In addition and similar to common practice in SSL-based image classification evaluation in the natural image domain context [19,29], a linear classifier (LL) was built by adding a fully connected (FC) layer (input: DINO representation, size: 384; output size: number of classes of the image data set) on top of the DINO backbone. The weights of the FC layer were trained by minimizing cross entropy (optimizer: Adam) based on augmented global crops of the input images, with the DINO-derived backbone weights $\theta_{t}$ being frozen. The stopping criterion was converging validation accuracy (relative improvement smaller than 0.5%), with the validation data set being a stratified split of 30% of the training set defined in Section 3.

### 2.4. Image Classification Using an In-House End-To-End Trained Supervised DL Baseline

For comparison purposes, we also built and trained an end-to-end trained supervised DL system. Regarding the network architecture, the model is identical to the LL classifier described in Section 2.3. Unlike training of the LL classifier, the DL baseline model initially consisted of the same ImageNet pre-trained XCiT model used for SSL in the DINO framework, and the weights of the pre-trained transformer were not frozen, but trained together with the FC layer.

### 2.5. Image Data Sets

Example images of the three different image data sets used in this study are shown in Figure 2.

#### 2.5.1. Bone Marrow (BM) Single Cell Data Set

The bone marrow (BM) data set published by Matek et al. [20] consists of 171,374 cropped microscopic cytological single cell microscopy images extracted from bone marrow smears from 945 patients with hematological diseases. The cell image size is 250×250 pixel, and the data set consists of 21 unbalanced classes with largely varying class frequencies (cf. Figure 4), reflecting the varying prevalence of disease entities and cell classes. The three largest classes (segmented neutrophils, erythroblasts, lymphocytes) each comprise more than 25,000 images. In contrast, the three smallest classes (abnormal eosinophils, smudge cells, Faggot cells) are represented by less than 50 images. Examples for the eight largest classes are shown in Figure 2.

#### 2.5.2. Endoscopic (Endo) Image Data Set

The Endo data set corresponds to the so-called HyperKvasir image data set published by Borgli et al. [21]). It contains 110,079 images captured during gastro- and colonoscopy examinations, of which 10,662 images are labeled. In turn, 99,417 images are not assigned with a label, making the HyperKvasir image data set a good candidate for the analysis of the potential of self-supervision for medical image classification. The labeled image data subset consists of images from 23 classes, including not only pathological findings but also anatomical landmarks (all classes: Barrett’s, bbps-0–1, bbps-2–3, dyed lifted polyps, dyed resection margins, hemorrhoids, ileum, impacted stool, normal cecum, normal pylorus, normal Z-line, oesophagitis-a, oesophagitis-b–d, polyp, retroflex rectum, retroflex stomach, short segment Barrett’s, ulcerative colitis grade 0–1, ulcerative colitis grade 1–2, ulcerative colitis grade 2–3, ulcerative colitis grade 1, ulcerative colitis grade 2, ulcerative colitis grade 3). Similar to the BM data set, the Endo data set is highly imbalanced, with the largest classes covering more than 1000 and the smallest classes less than ten labeled images. The size of the images varies between 332 × 487 and 1920 × 1072 pixel. Example images for the largest classes are shown in Figure 2.

#### 2.5.3. ISIC 2019 and 2020

The dermoscopic lesion data set is a collection of dermoscopic skin lesion data sets that were released as part of the annual Grand Challenges organized by the International Skin Lesion Collaboration (ISIC). For the present study, we used the ISIC 2019 (25,331 images) and the ISIC 2020 challenge data (33,126 images) [1,22,23,24]. The ISIC 2019 data set contains skin lesion images from eight different classes (actinic keratosis, basal cell carcinoma, benign keratosis, dermatofibroma, melanome, nevus, squamous cell carcinoma, vascular lesion; largest class: nevus, 12,875 images; smallest class: vascular lesion, 253 samples), see examples in Figure 2. The ISIC 2020 data set comprises only two image labels: melanoma and benign lesions [26]. The ISIC image size varies between 600 × 450 and 1024 × 1024 pixel. Our study focused on the more challenging multi-class setting, i.e., the ISIC 2019 setting. Therefore, the ISIC 2019 data were used as labeled data. However, compared to the BM and the Endo data set, the total number of ISIC 2019 images is relatively small for self-supervised representation learning. We, therefore, employed the ISIC 2020 image data as an additional unlabeled data pool for SSL.

**Figure 2 bioengineering-10-00895-f002:**
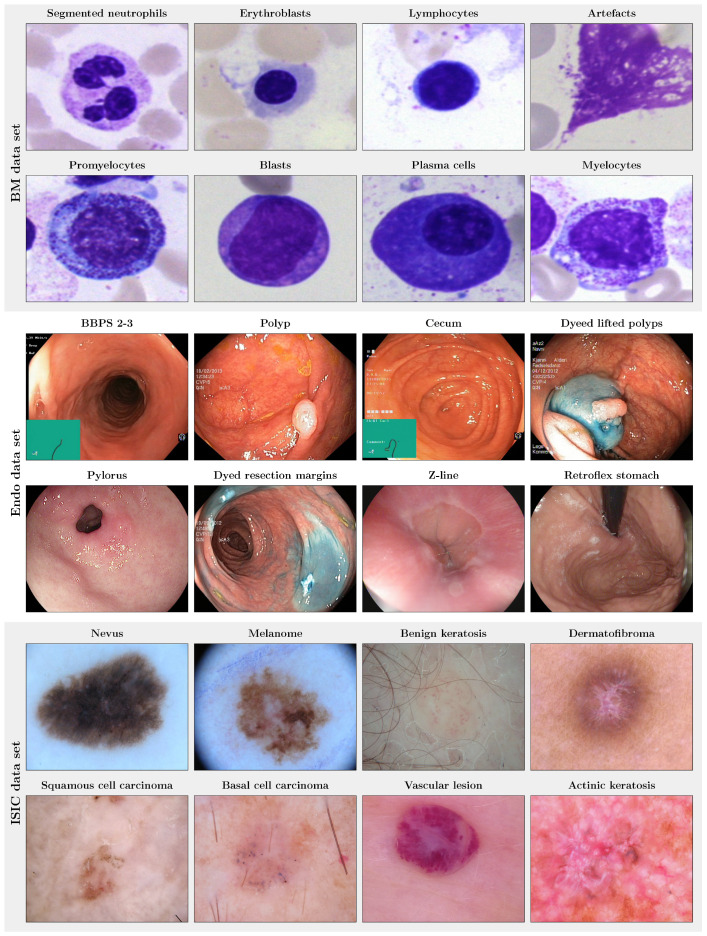
Example images of the three publicly available medical image data sets used in this study. (**Top panel**): images of the eight largest of the 21 classes of the bone marrow (BM) data set [20]. (**Mid panel**): images of the eight largest of the 23 classes of the endoscopic (Endo) data set, i.e., the HyperKvasir image data set [21]. (**Bottom panel**): images of the eight classes of the ISIC 2019 data set [22,24,25], i.e., dermoscopic images.

### 2.6. Experimental Setup

For each data set, the available labeled images were split into a train and a test set (70%/30%; stratified split to ensure the same class distribution in both sets). The entire training set, plus additional unlabeled data if available (see Section 2.5), was used to train an image-encoder backbone (XCiT) in a self-supervised fashion using the described DINO framework. In the next step, conventional ML classifiers as well as a FC layer (see Section 2.3) were fitted using the learned representation of a limited number of samples. Moreover, as described in Section 2.4, an XCiT model (architecture identical to the SSL backbone) was trained end-to-end in a supervised manner and serves as a supervised DL baseline.

For all approaches and data sets, the number of the labeled training samples per class was systematically varied between 1 and 1000 (steps: 1, 5, 10, 25, 50, 100, 250, 500, 1000 samples per class). In addition, two sets of experiments were performed: (1) experiments based on the entire set of classes provided as part of the public data sets, and (2) experiments based only on the classes for which at least 250 labeled training samples were available. Experiment series (2) aimed to provide an evaluation for a setting that was not affected by the pronounced class imbalance inherent in the full data sets. Finally, as an additional benchmark, the supervised model was also trained using all available labeled samples of the training data set.

For each experiment, the classification performance was evaluated using the modality-specific test sets. Due to long training times for the SSL training (exceeding multiple days for each data set; GPU: NVIDIA A40), only one model was trained per data set. Based on this model, each SSL experiment was repeated 100 times with different randomly sampled training splits. Due to also longer training times (e.g., 8–10 h for the BM data set with 250 samples per class), the supervised DL baseline experiments were repeated only five times, and 30% of the labeled training samples were used as validation data.

## 3. Results

The hypothesis of this study was that the image representations learned within the DINO framework, that is, without the use of labels, would allow accurate subsequent image classification with conventional ML classifiers and, compared to standard end-to-end deep learning, only a limited number of annotated image data.

### 3.1. Full Data Set Experiments and Effect of the Number of Labeled Training Samples

The first experiment series aimed at a direct comparison with literature values and covered all classes available in the labeled data sets. For the BM and the Endo data set, balanced accuracy values as reported in the original data set publications [20,21] were considered as literature benchmark and state-of-the-art performance. For the ISIC data set, the ISIC 2019 methods paper of the challenge winners Gessert et al. [30] was used as literature benchmark. All benchmark values were obtained through supervised end-to-end trained DL, using the maximum available number of labeled data.

The results for the three image modalities are, in terms of balanced accuracy, summarized in Figure 3. Figure 3 also shows the literature benchmark balanced accuracy.

Not surprisingly, improved accuracy and decreased variability of the individual runs were observed for an increasing number of training samples. However, the accuracy for the SSL-based approaches starts saturating already at a limited number of labeled samples per class for all data sets, with the exact number depending on the data set: For the Endo data set, a saturation at approximately 25 samples can be seen; for the BM data set, the accuracy converges around 50–100 samples per class; for the ISIC data set, saturation starts at ≥500 samples; the exact numbers depend on the classifiers. The later saturation for the ISIC data set can only partly be associated with the smaller number of classes and, consequently, the smaller number of absolute samples. The task itself seems to be more difficult to solve than the other two tasks, which is also reflected in the literature benchmarks given: The literature benchmark accuracy is at about the same level as for the other two tasks, although fewer classes have to be differentiated. As for the different classifiers, SVM-based classification seems to have a slight advantage over LR-, KNN- and LL-based classification for limited labeled data scenarios. The performance depends, however, on the data set.

**Figure 3 bioengineering-10-00895-f003:**
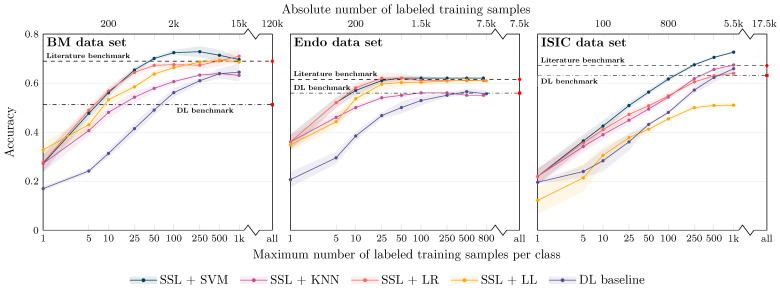
Classification balanced accuracy for the different classifiers for different numbers of labeled samples per class for the three image data sets (BM: bone marrow; Endo: endoscopic images; ISIC: dermoscopic images). The accuracy is shown for the *maximum* number of labeled training samples per class; if the training data set for a particular class contained only a smaller number of images, all available training images were used. This means that the training for larger sample sizes is partially biased toward better performance for larger classes due to the unavailability of a sufficient number of labeled training samples for smaller classes. As detailed in the main text, the literature benchmark performance refers to accuracy data reported in the context of the publication of the image data sets (BM, Endo) and a directly associated publication for ISIC ([30]; average accuracy of the 14 models evaluated in the paper). The DL benchmark line indicates the performance of our DL baseline model trained on all available labeled samples of the training data sets.

For all data sets, the literature benchmark data were reported as cross-validation (CV) results on a validation set, with the CV performed to determine the best model and hyperparameters; no independent test set was used. The values could therefore potentially be overoptimistic and subject to overfitting. In contrast, our DL baseline models were evaluated on the unseen test set. In line with the overfitting hypothesis, the test set accuracy for our DL baseline when trained using all available labeled samples of the training data set (referred to as DL benchmark in Figure 3) was consistently lower than the literature benchmark values for the three application examples. The validation set accuracy was, however, very similar to the reported literature accuracy.

Independent of this aspect: The proposed approach of self-supervised representation learning and subsequent image classification by conventional ML classifiers breaks the baseline and the literature benchmark lines with between 5 and 10 (our DL benchmark) and 25 and 50 labeled samples per class (literature benchmark) for the BM and the Endo data sets and with between 100 and 250 samples for the ISIC data set (classifier: SVM; see corresponding quantitative data in Table 1).

On the contrary, the balanced accuracy of our DL baseline, when trained with limited labeled image data, did not exceed the literature benchmark line for any of our limited label data scenarios, i.e., with less than 1000 samples per class. In addition, for small sample size scenarios, a clear performance gap compared to SSL with subsequent ML classification is visible in Figure 3. For example, for classification using 100 labeled samples per class, the accuracy for SSL and subsequent SVM classification is between 9% and 17% higher than the accuracy of the DL baseline. This demonstrates the ability of self-supervised DL methods and the proposed approach to achieve strong performance with a minimum of labeled samples.

For further illustration, Figure 4 shows the confusion matrix (CFM) for the BM data set and SVM-based classification with 100 labeled training samples per class, compared to the literature benchmark CFM diagonal as reported in [20]. While the overall balanced accuracy is similar for the two approaches (ours: 0.73; Matek et al.: 0.69), SSL leads to higher accuracy values for the smaller classes (balanced accuracy for the 50% smallest classes: 0.74 vs. 0.64). In turn, the supervised DL literature benchmark has a slight advantage for the larger classes (balanced accuracy for the 50% largest classes: 0.71 vs. 0.73). However, our results are based on only roughly 1% of the labeled data that were used for training the supervised DL benchmark. Using the same amount of labeled data for training of our DL baseline leads to a balanced accuracy of 0.56 and worse performance for all classes.

Similar observations also apply for the other two data sets, but SSL for the Endo and ISIC data require about 10% of the available labeled data to achieve performance comparable to the literature benchmark values.

### 3.2. Experiments with Classes with 250+ Samples

Figure 3 further shows that the performance of the ML classifiers only marginally gains from more than 250 samples per class. For some data set/classifier combinations (e.g., BM and SVM for more than 100 samples per class, or Endo and KNN for more than 250 samples per class) and the full data set setting (i.e., consideration of all classes), the accuracy even declines if more labeled training samples per class are used to fit the classifier parameters. This behavior is only due to a drop in performance for smaller classes and can be hypothesized to be a consequence of the pronounced class imbalance in the data sets, which is exacerbated in scenarios with a larger number of labeled training samples. For the three smallest classes of the BM data set, for instance, the number of available labeled training images was 6 (abnormal eosinophils), 29 (smudge cells), and 33 (faggot cells). For the experiments and results described so far, we accounted for class imbalance using the standard approaches for the ML classifiers (SVM, LR: scaling of class weights inversely to the class sizes; KNN: weighting of data points inversely to the distances to the query point). The second experiment series aimed to obtain a sound performance analysis of the capabilities of self-supervised representation learning independent of potential issues due to class imbalance (see Section 2.6); therefore, we repeated the experiments, i.e., classifier training and evaluation, using only the classes that cover more than 250 samples in the training data set. For clarification purposes, the corresponding data sets are subsequently denoted by an appended 250+ (e.g., *BM250+*).

**Figure 4 bioengineering-10-00895-f004:**
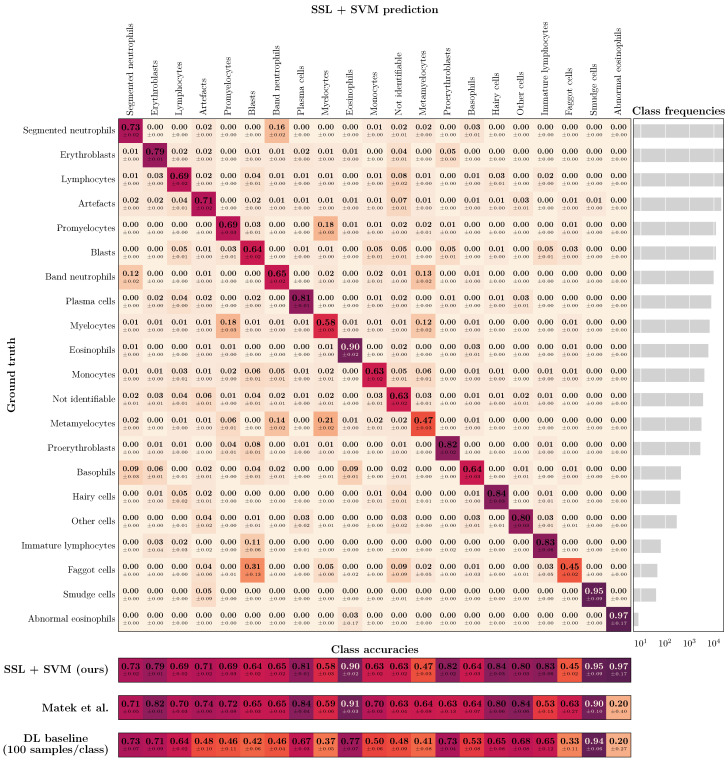
(**Top**): Confusion matrix for bone marrow (BM) image classification using the proposed workflow (i.e., SSL and subsequent classification with a standard ML classifier) with SVM training based on only 100 labeled images per class. The class frequencies are shown on the right. (**Bottom**): Comparison of diagonal elements of the proposed approach (SSL + SVM), the benchmark data reported by Matek et al. [20], and our DL baseline model when trained on only 100 labeled images per class. Please note that it is possible to zoom into the digital version of the figure for full readability of the details.

The results are shown in Figure 5. As hypothesized, different from the full data set experiments, no drop in performance for an increasing number of samples can be seen for the 250+ data sets, and the general trend is similar as for the full data set experiments. The literature benchmark line in Figure 5 now refers to the accuracy data reported for the classes contained in our 250+ data set. For the BM and Endo data sets, again, the proposed SSL approach achieves similar performance with fewer samples. The specific benchmark information was unfortunately not available for the ISIC data set.

**Figure 5 bioengineering-10-00895-f005:**
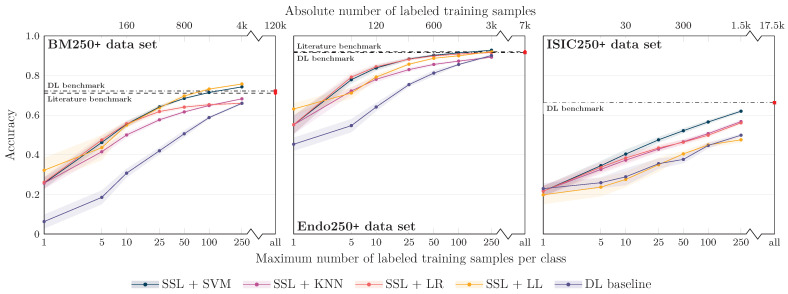
Classification balanced accuracy for the different classifiers for different numbers of labeled samples per class for the three image data sets, similar to Figure 3, but the experiments were restricted to classes with more than 250 labeled training samples (BM data set: 16 classes; Endo: 12; ISIC: 5) to avoid bias of the classifier training toward large classes. In experiments with the number of training samples per class of more than 250, again, a class imbalance would have biased the results. We, therefore, focused on experiments with fewer than 250 samples per class, although it is obvious for the ISIC data set that both the SSL and the DL approaches would gain in performance with more labeled samples.

The comparison to the literature values is, however, biased because no explicit supervised DL model was trained by the authors to differentiate only the 250+ classes. To conquer the bias, we also re-trained and evaluated our DL baseline models on the 250+ data sets. Trained on all available labeled samples of the 250+ classes, the accuracy of DL baseline is now similar to the literature benchmark values. Again, different from the literature values, the in-house DL benchmark accuracy data were obtained based on a sound test data set. Training the DL baseline models with limited labeled image data also supports the results for the full data set experiments: Although the extent of the effect depends on the specific data set, a clear gap between the SSL-based classification approaches and the DL baseline models is evident for all data sets for small sample sizes.

Compared to the results for the full data sets, interestingly, the saturation of the performance curves starts with a higher number of samples per class. As we reduced the overall number of classes, this indicates that not only the relative amount of samples (i.e., samples per class) is important but still also the absolute number of samples. This might again be a potential explanation for the slower increase in performance for the ISIC data set.

## 4. Discussion

The present study demonstrates that self-supervised DL feature extraction combined with conventional ML classification is capable of achieving competitive performance in medical image classification even under very limited labeled data availability conditions: Using the DINO framework [19], we achieved state-of-the-art classification performance on three different public medical data sets of high interest in the community making use of only 1% (BM data set) and 10% (Endo and ISIC data sets) of the available labeled data and approximately 100 labeled samples per class (between 25 and 250, depending on the data set), respectively. In contrast, corresponding literature benchmark approaches are based on supervised DL making use of all available labeled image data. When interpreting the results, it should further be noted that the literature benchmark performance refers to results for validation data sets (and not test data, as in our study), which tends to overestimate model performance on a test data set. This hypothesis was supported by the weaker performance of the baseline DL models that we trained in a supervised end-to-end approach using the entire labeled training sets and a clean train/test split. Our results therefore highlight the capability of self-supervision for medical data analysis, a field, in which data annotation is time consuming, expensive in terms of expert work load, and eventually a limiting factor for standard supervised DL.

Similar to the computer vision community, there is currently a trend toward releasing large(r) publicly available annotated medical image data sets [31]. It can be argued that with the availability of such data sets, the pressure and need to develop label-efficient learning approaches will be eased. We do not think so. The data sets usually come with only a single set of labels, tailored to a specific scientific or clinical question. An adjustment of the specific question require partial or full relabeling of the entire data set.

Furthermore, end-to-end trained DL systems usually need to be re-trained (including hyperparameter optimization) after adaptation of the specific research question and corresponding data relabeling. This is a time-consuming and power-intensive process. In contrast, SSL-based image representations can be directly used for rephrased research questions and downstream tasks. For image classification, the use of standard ML approaches facilitates efficient (i.e., fast, smaller energy consumption) reuse of the SSL features.

At the same time, large data sets hold promise to unlock the full potential of SSL for medical image analysis, since they cover most common clinical image domains. This opens the door for efficient combination of public and private (and potentially labeled) image data sets to tackle specific task and allows researchers who have only access to limited (labeled) data to effortless enrich their data and to contribute to method development and medical data analysis.

The demonstrated promising performance of self-supervision for medical image classification now suggests extending the present study as follows:

First, we focused on two-dimensional image data sets. Typical radiological images such as computed tomography or magnetic resonance imaging data are usually volume data, i.e., three-dimensional data were not included. In addition, temporally resolved image data representing either physiological processes or imaging follow-up data add another dimension. The potential of self-supervised representation learning for these data types has still to be explored in detail.

Second, the present study addressed a multi-class and whole image classification setting. Intrinsically, the applied DINO self-supervision framework forces similarity between global and local crops of the same image, which can be assumed to be well suited for global, i.e., whole image labels. Especially for volumetric medical images and multi label settings which are also common *meaningful* representations would also on local changes and/or multiple pathological alterations in a single image/volume. It remains to be analyzed whether the learned DINO representations are also capable to achieve state-of-the-art performance with limited labeled data for these settings.

Third, in line with the second aspect, it will be interesting to explore the capabilities of different self-supervision approaches like contrastive learning [16,29,32] and, if existing, to identify well-suited task-specific self-supervision methods. Similarly, a better understanding of the impact of different components of SSL frameworks on their performance (e.g., by corresponding ablation studies) would potentially help to further improve them. For instance, for the used DINO framework, this applies the data augmentation approaches, which are currently adapted from the natural image domain and are not necessarily optimal for the medical image domain or specific subdomains.

Thus, we will see to which degree the ongoing theoretical and methodical developments in this field allow an additional performance gain, whether corresponding models will be taken up more frequently by the community, and whether this eventually allows the clinical domain experts to spend their valuable time for more useful activities than image annotation. We hope that the present publication and the corresponding results will be taken up as benchmark data in the field of SSL-based medical image data classification. The models, data splits and results are publicly provided to ensure reproducibility.

## Figures and Tables

**Table 1 bioengineering-10-00895-t001:** Balanced accuracy for medical image classification using self-supervised image representation learning and subsequent SVM classification for different numbers *n* of labeled samples per class (quantitative data, corresponding to Figure 3 and Figure 5). The DL benchmark values (DL) refer to model training with the all available labeled training samples and the benchmark performance to literature values reported in the context of the publication of the image data sets (see text for details). Best accuracy values for the different data sets are highlighted by bold text.

	All Classes	Classes ≥250 Samples
*n*	BM	Endo	ISIC	BM250+	Endo250+	ISIC250+
1	0.27_(0.04)_	0.36_(0.03)_	0.22_(0.03)_	0.26_(0.03)_	0.55_(0.05)_	0.22_(0.03)_
5	0.48_(0.02)_	0.52_(0.02)_	0.36_(0.03)_	0.46_(0.02)_	0.78_(0.02)_	0.34_(0.02)_
10	0.56_(0.02)_	0.57_(0.02)_	0.42_(0.02)_	0.55_(0.01)_	0.84_(0.01)_	0.40_(0.01)_
25	0.65_(0.01)_	0.61_(0.01)_	0.51_(0.01)_	0.64_(0.01)_	0.88_(0.00)_	0.48_(0.01)_
50	0.70_(0.01)_	**0.62** _(0.01)_	0.56_(0.01)_	0.68_(0.00)_	0.90_(0.00)_	0.52_(0.01)_
100	0.72_(0.01)_	0.62_(0.01)_	0.62_(0.01)_	0.71_(0.00)_	0.91_(0.00)_	0.57_(0.01)_
250	**0.73** _(0.02)_	0.62_(0.00)_	0.67_(0.00)_	**0.74** _(0.00)_	**0.93** _(0.00)_	**0.62** _(0.01)_
500	0.71_(0.02)_	0.62_(0.00)_	0.70_(0.00)_			
1000	0.70_(0.01)_	0.62_(0.00)_	**0.73** _(0.00)_			
DL^  ^	0.51	0.56	0.63	0.72	0.92	0.67
Lit.^ 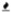 ^	0.69	0.62	0.67	0.71	0.92	n/a

^

^ In-house DL benchmark, trained on all labeled training samples, evaluated on test set. ^
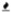
^ Literature benchmark, trained on all labeled training samples, evaluated with CV.

## Data Availability

All three data-sets used in this study are publicly available. The ISIC [22,23,24,25,26] dataset is available through the following link: https://challenge.isic-archive.com/data/ (accessed on 23 July 2023). The data-set of bone marrow cells affiliated with the publication “Highly accurate differentiation of bone marrow cell morphologies using deep neural networks on a large image data set” [20] is available through the cancer imaging archive: https://wiki.cancerimagingarchive.net/pages/viewpage.action?pageId=101941770 (accessed on 23 July 2023). The data-set of endoscopic intervention (HyperKvasir) [21] may be found under the following DOI: https://doi.org/10.6084/m9.figshare.12759833.v1 (accessed on 23 July 2023), https://doi.org/10.17605/OSF.IO/MH9SJ (accessed on 23 July 2023). All code used in context of this work is available under https://github.com/IPMI-ICNS-UKE/sparsam (accessed on 23 July 2023).

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
