# Peer review of "Self-Supervision for Medical Image Classification: State-of-the-Art Performance with ~100 Labeled Training Samples per Class"

_bioengineering, 2023, doi:10.3390/bioengineering10080895_

Round 1

Reviewer 1 Report

This manuscript evaluates the use of the self-distillation with no labels (DINO) framework combined with conventional machine learning classifiers for medical image classification on three medical imaging dataset (bone marrow microscopy, gastrointestinal endoscopy, dermoscopy). They achieve SOTA classification performances using only about 100 labeled images per class.

The manuscript is well written. The aim is clear and the results support the conclusions. My main concern with this manuscript is that it is not very novel or innovative: the authors merely applied the DINO implementations to medical data. In addition, before this manuscript may be considered for publication, I recommend the authors address the following points:

-        It would be interesting to see how much each aspect of the DINO framework (crops, different types of augmentations, etc.) contributes to the performance gain.

-        The ISIC dataset shows different curves/trends compared to the other datasets, why?

-        There is some repetition in the discussion.

-        The Methods section is located in a strange place (at end of the manuscript), this is not helpful for the flow of the information in the manuscript.

-        A graphical representation of the network architectures (XCiT + ML classifiers and the DINO framework) would be helpful.

-        Some explanations in the Methods section are a bit difficult to follow for the audience of Bioengineering, particularly sec. 4.3 about how the output of the trained DINO backbone was used to train the ML classifiers.

-        More information on alternative SSL techniques would be helpful, including corresponding references. Why did the authors choose to focus on the DINO framework, besides the fact that this framework was introduced recently? Is it possible to include a benchmark comparison with another SSL framework?

Reviewer 2 Report

This article highlights the investigation of self-supervised DL as an alternative to the standard supervised DL approach for medical image analysis, specifically focusing on the challenge of limited availability of labeled data. The study evaluates the performance of self-supervised DL using the self-distillation with no labels (DINO) framework on three common medical imaging datasets: bone marrow microscopy, gastrointestinal endoscopy, and dermoscopy. The authors employ conventional machine learning classifiers after learning image representations without using image labels. They systematically vary the number of labeled data samples (ranging from 1 to 1000 per class) and assess the classification performance. The results demonstrate that the self-supervised DL approach achieves state-of-the-art classification performance for all three imaging datasets using only a small fraction of the available labeled data. The approach appears to effectively address the issue of limited labeled data, offering improved classification performance with a reduced amount of labeled data. 

Some minors:

(1) The foundational concepts and theoretical underpinnings of self-supervised learning are inadequately presented.

(2)The text within the confusion matrix depicted in Figure 3 presents readability challenges. To enhance legibility, it is recommended to enlarge the font size within the figure.

(3) The font size of the label in the middle sub-diagram of Figure 4 appears to be inadequately small, please enlarge the font size of the label in question.

Round 2

Reviewer 1 Report

I have no further comments. It would have been nice to include a comparison with other SSL methods in this work, but I understand that time in rebuttal is limited. The study itself is interesting and the amendments made by the author have substantially improved the readability of the manuscript. Well done!